# Epidemiological Impact on Use of Antibiotics in Patients Hospitalized for COVID-19: A Retrospective Cohort Study in Italy

**DOI:** 10.3390/antibiotics12050912

**Published:** 2023-05-15

**Authors:** Zaira Maraia, Tony Mazzoni, Miriana Pia Turtora, Alessandra Tempera, Marco Spinosi, Anita Vagnoni, Isidoro Mazzoni

**Affiliations:** 1School of Specialization in Clinical Pharmacology and Toxicology, University of L’Aquila, 67100 L’Aquila, Italy; zairamaraia@gmail.com; 2School of Specialization in Hospital Pharmacy, University of Camerino, 62032 Camerino, Italy; tony.mazzoni97@gmail.com (T.M.); mirianapiaturtora@gmail.com (M.P.T.); ale_temp@hotmail.it (A.T.); 3Ascoli Piceno Hospital Pharmacy, 63100 Ascoli Piceno, Italy; marco.spinosi@sanita.marche.it (M.S.); anita.vagnoni@sanita.marche.it (A.V.)

**Keywords:** antibiotic use, coinfections, pandemics, SARS-CoV-2

## Abstract

The increased incidence of antimicrobial resistance during coronavirus disease 2019 (COVID-19) is a very important collateral damage of global concern. The cause is multifactorial and is particularly related to the high rates of antibiotic use in COVID-19 patients with a relatively low rate of secondary co-infection. To this end, we conducted a retrospective observational study of 1269 COVID-19 patients admitted during the years 2020, 2021 and 2022 in two Italian hospitals, with a focus on bacterial co-infections and antimicrobial therapy. Multivariate logistic regression was used to analyze the association between bacterial co-infection, antibiotic use and hospital death after adjustment for age and comorbidity. Bacterial co-infection was detected in 185 patients. The overall mortality rate was 25% (*n* = 317). Concomitant bacterial infections were associated with increased hospital mortality (β = 1.002, *p* < 0.001). A total of 83.7% (*n* = 1062) of patients received antibiotic therapy, but only 14.6% of these patients had an obvious source of bacterial infection. There was a significantly higher rate of hospital mortality in patients who received antibiotics than in those who did not (χ^2^ = 6.22, *p* = 0.012). Appropriate prescribing and the rational use of antimicrobials according to the principles of antimicrobial stewardship can help prevent the emergence of antibiotic resistance.

## 1. Introduction

Severe acute respiratory syndrome coronavirus 2 (SARS-CoV-2) has caused a sudden and substantial increase in hospitalizations for pneumonia worldwide, eventually progressing to hypoxic respiratory failure, respiratory distress syndrome (ARDS) and multi-organ failure [1,2]. Now, the effect of COVID-19 disease on susceptibility to bacterial infections is still not well understood and has therefore been raised as an important knowledge gap in our country. Viral pathogens, such as the influenza virus and rhinovirus, have been shown to predispose patients to bacterial infections and subsequent increased mortality [3]. Direct damage of the mucosa by the virus, increased bacterial colonization of the airways and dysregulation of immune responses all lead to increased susceptibility to secondary bacterial infections [4]. 

Bacterial co-infection in SARS-CoV-2 has been the focus of attention, as patients with COVID-19 may be vulnerable to other infections due to multiple comorbidities, prolonged hospitalization and immune dysfunction. However, the prevalence of co-infection was remarkably low. 

A meta-analysis conducted by Langford et al. showed that 7% of patients with COVID-19 acquired a secondary bacterial infection [5]. In Spain, an observational cohort study involving 989 patients with SARS-CoV-2 infection showed that 7.2% of patients had a microbiologically confirmed infection and 3.1% had acquired co-infection in the community at the diagnosis of COVID-19 [6]. In the Netherlands, a retrospective observational study showed that only twelve (1.2%) of the 925 included patients had a documented bacterial co-infection [7]. A review by Westblade et al. showed that in almost all studies analyzed, less than 4% of hospitalized patients had a documented bacterial co-infection [8]. In another meta-analysis, 7% of hospitalized COVID-19 patients had a bacterial co-infection, showing an increase to 14% in studies that included only intensive care patients [9]. 

Thus, the literature shows that bacterial co-infections are less prevalent in COVID-19 patients than in influenza patients. The World Health Organization (WHO) guidelines for the clinical management of COVID-19 advise doctors to collect blood cultures and respiratory specimens from the upper respiratory tract for bacterial cultures and to initiate empirical antimicrobial treatment only in severe cases [10].

Despite much evidence, it was observed that the rate of antibiotic use was significantly higher than the incidence of confirmed secondary infections in COVID-19 patients. An overuse of antibiotics increases the risk of multi-resistant nosocomial secondary infections associated with unfavorable clinical outcomes [11,12]. 

In this regard, a recent prospective study evaluating the impact of antibiotic treatment on the outcomes of hospitalized patients with moderate to severe COVID-19 had shown that patients who received antibiotics during hospitalization had a higher mortality (RR = 3.37, 95% CI: 1.7–6.8), and this association was stronger in the subgroup of patients without reasons for antimicrobial treatment (RR = 6.1, 95% CI: 1.9–19.1), whereas in the subgroup with reasons for antimicrobial therapy, the association was not statistically significant (OR = 2.33, 95% CI: 0.76–7.17) [13]. 

Therefore, the practice of empirical antibiotic coverage in COVID-19 patients must be carefully considered. Prescribing by physicians and dispensing by pharmacists should be done ethically and with scrupulous judgment.

### Increased Incidence of Antibiotic Resistance in the COVID-19 Era

Antibiotic resistance refers to the adaptation of a bacterium to the selective pressure exerted by a given antibiotic resulting in drug inefficiency, persistent infection and increased risk of severe disease. The emergence of resistance is inevitable and not a new phenomenon. In other words, it is the use of the antibiotic itself that increases the likelihood of such adaptation taking place [14]. Antibiotic resistance represents one of the most important threats to global health, and in the absence of new drugs, it is estimated that bacterial infections alone could cause hundreds of millions of deaths [15]. The inappropriate prescription of antibiotics and over-the-counter sales of antibiotics are some of the factors driving antimicrobial resistance globally [16]. During the COVID-19 pandemic, there was an increase in the use of antimicrobial agents compared to previous years [17,18,19]. This scenario is expected to have seriously facilitated the growth rate of antimicrobial resistance worldwide. In this respect, studies have already described an increase in resistant bacteria during the COVID-19 pandemic. 

An Italian retrospective study found an increase in the incidence of colonization by carbapenemase-resistant *Enterobacterales* from 6.7% in 2019 to 50% in 2020 [20]. Li et al. showed that among 159 strains of bacteria isolated from 102 hospitalized COVID-19 patients, *Acinetobacter baumannii* and *Klebsiella pneumoniae* were the main bacteria. The carbapenemes resistance rates for the bacteria just mentioned were 91.2% and 75.5%, respectively. Methicillin resistance was present in 100% of *Staphylococcus aureus* [21]. Another French retrospective study found that 26 COVID-19 patients admitted to the intensive care unit acquired bacterial co-infection, and 21% of these were resistant to third-generation cephalosporins and amoxicillin/clavulanate [22]. 

In this direction, updating the epidemiology of resistant pathogens to make rational therapeutic decisions becomes essential. Patients with COVID-19 are given empirical antibiotic therapy for possible bacterial pneumonia. However, the available evidence on co-infections in the COVID-19 context does not support the routine use of empirical antibiotics. The WHO is also extremely concerned about this. A new WHO report published on 12 December 2022 shows high levels (above 50%) of resistance in bacteria that frequently cause blood infections in hospitals, such as *Klebsiella pneumoniae* and *Acinetobacter*. Over 60% of *Neisseria gonorrhea* cases showed resistance to one of the most widely used oral antibacterial, ciprofloxacin. Over 20% of *Escherichia coli* cases are resistant to both first-line drugs (ampicillin and trimoxazole) and second-line treatments (fluoroquinolones) [23]. 

Although some of the resistance trends have remained stable over the past four years, the situation is not crystallized, as the use of antibiotics is the main determinant of this phenomenon.

This study aims to determine the incidence of bacterial co-infection in hospitalized patients with COVID-19 by analyzing the antibiotic therapies used in our cohort to expand existing knowledge. Furthermore, it wants to evaluate the impact of antibiotic treatment on the outcome (death/survival) of patients hospitalized with COVID-19. Understanding the proportion of COVID-19 patients with bacterial co-infection is crucial to ensure the responsible use of antibiotics and to minimize the negative consequences of antibiotic resistance.

## 2. Results

A total of 1269 patients tested positive and were admitted for COVID-19 at the hospital clinic in Ascoli Piceno and San Benedetto del Tronto in 2020, 2021, and 2022. 

Table 1 shows the general and clinical characteristics of the sample under analysis. 

The mean age ± SD of the total sample was 71.3 ± 2.5. More than half of the patients were male (56.1%; *n* = 712). The comorbidities present included diabetes (*n* = 207), hypertension (*n* = 173), atrial fibrillation (*n* = 115), ischemic heart disease (*n* = 81), obesity (*n* = 78), heart failure (*n* = 70), renal failure (*n* = 58), cancer pathology (*n* = 51), chronic obstructive pulmonary disease (COPD) (*n* = 40) and asthma (*n* = 11).

The total amount of patients hospitalized for symptomatic COVID-19 was 15.3% (*n* = 194), those for pneumonia comprised 22.3% (*n* = 283) and those for ARDS comprised 62.4% (*n* = 792). Although admissions for ARDS represent the majority in the sample as a whole (Table 1), compared to 2020 and 2021, admissions for ARDS and pneumonia decrease significantly in 2022 and admissions for mild/moderate symptomatic COVID-19 increased. Admissions to intensive care decreased from 25.2% during 2020–2021 to 0.3% in 2022 in a statistically significant manner (*p* < 0.001). The mean ± SD of total hospitalization was 13.8 ± 2.5, while the overall mortality was 25% (*n* = 317).

The concomitant documented bacterial infection rate was 14.6% (*n* = 185); 65.3% of infections were detected on admission, and 34.7% were detected during hospitalization.

Figure 1 shows the rate of bacterial infections in the years 2020, 2021 and 2022. Respiratory infections were the most common (51.1%), followed by urinary (37.3%), gastrointestinal (3.3%) and cardiac (2%) infections.

Subdividing the cohort according to clinical outcome revealed that bacterial co-infection was found in 23.8% of patients with mild COVID-19, 27.2% of patients with COVID-19 pneumonia and 49.2% of patients with COVID-19 ARDS. In addition, 83.2% of patients with co-infection (*n* = 154) were admitted to the general ward and 16.7% (*n* = 31) were admitted to the intensive care unit. Comorbidity was present in 57.3% (*n* = 106) of patients with co-infection.

The analysis revealed a significantly higher rate of in-hospital mortality in patients with concomitant bacterial infection than in those without (χ^2^ = 37.41, df = 1, *p* < 0.001). The death rate among those without co-infection was 21.9% and increased to 43.2% in the presence of co-infection (Figure 2). The presence of co-infection leads to a relative risk of 43.2/21.9 = 1.97.

A logistic regression model was constructed to explain the probability of death as a function of the presence of co-infection. The model is significant, and the coefficient of the variable is 1.002 (*p* < 0.001); thus, the presence of co-infection increases the probability of death.

Furthermore, a comparison of the mean age of the patients showed a significant difference between −6.9 and −2.5 years (t = −4.135, df = 264.62, *p* = 0.03). The mean age of patients without co-infection was 70.6 years, whereas the mean age of patients with co-infection was 75.3 (Figure 3). Thus, patients with co-infection are older than those without.

A more comprehensive logit model (Table 2) was created which also considers the patient’s age, the presence of comorbidities, the severity of the illness and the admission department. The purpose of the logit model is to study which variables have the greatest influence by trying to interpret reality. When the β coefficients of the model are positive, this means that the probability of death increases; conversely, when they are negative, this means that the probability of death decreases. In our case, all the regression coefficients are positive and statistically significant. Including indices such as age, comorbidity, severity of illness, and intensive care unit admission, bacterial co-infection is associated with an increased probability of death (*p* = 0.000).

As shown in Figure 4, the bacterial co-infection rate was stable at 11% during the years 2020–2021 with an increase of 10% in 2022. 

In terms of antibiotic prescription patterns, 83.7% (*n* = 1062) of all patients received antibiotics, but only 14.6% (*n* = 185) of these patients had an obvious source of bacterial infection The highest use of antibiotics was in 2021. Death was the outcome in 317 patients: 280 in the antibiotic-treated group and 37 in the untreated group. No significant difference was observed between the two groups in terms of mean length of hospital stay: 13 days in the surviving patients compared to 11 days in the deceased patients.

Similarly, a higher mortality rate emerged in antibiotic-treated patients than in untreated ones (χ^2^ = 6.22, df = 1, *p* = 0.012). In fact, death occurred in 26.4% of treated patients compared to 17.9% of untreated patients (Figure 5). In addition to this, the comparison of the mean age did not reveal a significant difference, being between −4.4 and 0.8 (t = −1.3522, df = 259.39, *p* = 0.1775), showing a mean age of 69.7 years in the untreated patients compared to 71.5 years in the treated patients.

A more comprehensive logit model (Table 3) was created which also takes into account the patient’s age and the presence of comorbidities. The purpose of the logit model is to study which variables have the greatest influence by trying to interpret reality. When the β coefficients of the model are positive, this means that the probability of death increases, and conversely, when they are negative, this means that the probability of death decreases. All the variables examined have positive and statistically significant regression coefficients. Including indices such as age and comorbidity, antibiotic use is significantly associated with an increase in hospital mortality (β = 0.493, *p* = 0.019).

We wanted to investigate whether treatment with antibiotics was mainly used in severely ill patients (Figure 6). Comparison of the averages revealed a significant difference (t = −7.6777, df = 258.32, *p*-value = 0.000) between 2.1 and 2.5. Specifically, 11.4% of the antibiotic-treated patients had mild COVID-19, 22% had pneumonia and 66.6% had ARDS, whereas 23.7%, 35.3% and 41.1% of the untreated patients had these diagnoses, respectively. Thus, the analysis shows that patients treated with antibiotics had more severe and critical COVID-19 disease.

Exploratively, the most frequently used antibiotics in treatment were azithromycin (31.4%), piperacillin/tazobactam (28.6%), ceftriaxone (25.7%), levofloxacin (25.2%), meropenem (14.8%) and linezolid (11.7%) (Figure 7). In terms of the number of units of drug administered, the largest quantities were found with piperacillin/tazobactam and meropenem in 2022; while the largest quantities were found with levofloxacin, linezolid and vancomycin in 2021.

To investigate which of the most frequently used antibiotics were associated with a higher probability of death, an additional logistic model was constructed, also considering age and comorbidity (Table 4). The analysis showed that the use of piperacillin/tazobactam (β = 0.556, *p* = 0.001), meropenem (β = 0.766, *p* = 0.001) and linezolid (β = 0.903, *p* = 0.001) was statistically significantly associated with increased mortality.

Finally, the categorical variable ‘Empirical Antibiotic’ was constructed to identify patients who were administered the antibiotic in the absence of bacterial co-infection. In this subgroup, the number of patients who died was *n* = 202. As shown in Figure 8, the proportion of deaths among those who did not receive empirical antibiotics was 17.8% and increased to 22.8% when empirical antibiotics were used. However, the 5% increase found in the sample was not statistically significant (χ^2^ = 2.0815, df = 1, *p* = 0.1).

## 3. Discussion

In the present retrospective cohort study, a bacterial co-infection rate of 14.6% was found. In comparison to an Italian observational study of COVID-19 patients, only 6.3% had an underlying infection [24]. The higher co-infection rates in our cohort may be justified by a relatively more compromised patient population with a higher percentage of diabetes. 

Several theories highlight how cardiometabolic comorbidities, due to an underlying dysregulated immune response, lead patients to have higher rates of bacterial co-infection in viral infection. Indeed, diabetes mellitus itself by reducing the T-lymphocyte and neutrophil response causes a reduced innate immune response leading to increased susceptibility to secondary infections [25]. Patients with co-infection were relatively older than those without.

Mortality was observed in 43.2% of our COVID-19 patients with concomitant bacterial infection (χ^2^ = 37.41, *p* < 0.0001). We observed that bacterial infection was a predictor of increased mortality (β = 1.002, *p* < 0.001). The presence of co-infection could be linked to clinical deterioration rather than a cause of worse clinical outcomes. In fact, increasing age, the presence of comorbidities and the severity of the disease are also strong risk factor for death in hospital. Our patients with co-infections were more likely to die in the intensive care unit. Similar results emerged from a case-control study that investigated risk factors for bacterial infections in patients with moderate to severe COVID-19 and found statistical significance in the case of critical illness [26].

In our cohort, the bacterial co-infection rate was stable and 11% during the years 2020–2021 with a 10% increase in 2022. This could presumably be attributed to the reduction in anti-COVID-19 protective measures that resulted in an increase in bacterial infections [27].

Overall, 51% of our patients had a respiratory infection identified by sputum culture. As shown in previous reviews, the most common bacterial co-infections were genito-urinary tract infections, accounting for 57–70% of cases, which was followed by 19% respiratory tract infections, skin and soft tissue infections in 1% [28]. In our study, the rate of non-respiratory bacterial infections was 42%, but as the patients had a relatively long hospital stay, some of these were hospital-acquired.

In this study, more than half of the patients received antibiotics (*n* = 1069). Azithromycin, piperacillin/tazobactam and ceftriaxone were the most prescribed antibiotic regimens. This may reflect concerns about possible complications of the disease due to bacterial co-infections, as has been previously described in the case of influenza and other viral respiratory infections [29]. At first, azithromycin was also prescribed for its alleged immunomodulatory action, but later studies have shown that there is no improvement in survival or other clinical outcomes [30].

In our cohort, mortality was observed in 26.4% of patients treated with antibiotics (χ^2^ = 6.22, *p* = 0.012). We observed that antibiotic use is associated with increased mortality (β = 0.493, *p* = 0.019). However, in our study, patients who died and were treated with antibiotics had more comorbidities and had more severe forms of COVID-19.

The data that emerged from our study are broadly consistent with those reported in the literature.

A recent systematic review including 118 studies showed that 8501 out of 10,329 patients with COVID-19 were prescribed antibiotics. Furthermore, the results show that patient mortality was higher in patients treated with antibiotics than in untreated patients (26.5% vs. 2.3%). The percentage of patients who were prescribed antibiotics without clinical justification was 51.5% compared to 41.9% for patients with mild or moderate disease and those with severe or critical disease [31].

Another meta-analysis involving 24 studies with 3338 COVID-19 patients showed that 71.9% (95%CI 56.1–87.7%) of them received antibiotics, but the overall percentage of COVID-19 patients with bacterial infection was 6.9% (95% CI 4.3–9.5%) [5].

Similar results were reported in another meta-analysis that included 154 studies: the prevalence of antibiotic prescription was 74.6% (95% CI, 68.3–80.0%), which increased with patient age (OR, 1.45 per 10-year increase, 95% CI 1.18–1.77%) and the percentage of patients requiring mechanical ventilation (OR, 1.33 per 10% increase, 95% CI 1.15–1.54) [11].

Given the high rates of empirical antibiotic use without evidence of bacterial infection, appropriate antibiotic management in the context of COVID-19 remains a challenge. There are currently limited data on antibiotic use in the COVID-19 patient in Italy. A recent study demonstrated a substantial increase in hospital antibiotic consumption during the COVID-19 pandemic in Italy [32].

To avoid mortality related to the new coronavirus, suboptimal antimicrobial stewardship may have encouraged the emergence of further antibiotic resistance. For this reason, there is a growing interest in antimicrobial stewardship, i.e., a series of multidisciplinary interventions aimed at promoting the appropriate use of antibiotics in which the contribution of the hospital pharmacist is necessary for the optimal choice of drug, dose and duration of therapy [33]. The need to strengthen the entire system of management, control and monitoring within hospital facilities becomes more urgent every day as antibiotic resistance is a growing problem globally but with a significant impact especially in our country. Indeed, in Italy, infections caused by antibiotic-resistant bacteria cost the lives of over 10,000 people every year and rank first in Europe in terms of the number of deaths. The report on the use of antibiotics in Italy produced by the Italian Medicines Agency (AIFA) and published in January 2021 shows that the rate of prescriptions for this class of drugs is high [34]. Approximately 4 out of 10 citizens received at least one antibiotic prescription during the year, 80% of which were issued by the National Health Service. Concerning the use of antibiotics in hospital care, a consumption of 77.2 DDD/100 hospital days emerged in 2019. The three most widely used classes of antibiotics in this context appear to be penicillins associated with beta-lactamase inhibitors, third generation cephalosporins and fluoroquinolones [34]. In addition, according to a recent meta-analysis in Italy, out of 21,260 isolated strains, 9507 showed antibiotic resistance (45%), of which 3905 (18%) were β-lactams and tetracyclines resistant, followed by aminoglycosides and quinolones resistant [35]. Exacerbating the scenario is the lack of knowledge of Italian citizens, which shows that only 9.8% know the definition of antibiotic resistance correctly and 32.7% classify them as self-medication drugs to treat throat, fever and the common cold [36]. Patient education is crucial in the fight against antibiotic resistance.

Since the beginning of the pandemic, national authorities have strongly discouraged the use of antibiotics for COVID-19. As amply demonstrated by numerous well-conducted clinical studies, there is no evidence that the use of antibiotics has a protective effect on the evolution of COVID-19 including mortality, as clearly stated by all international guidelines for the treatment of SARS-CoV-2 infection [37,38,39,40]. Since antibiotics probably do not provide a benefit as an empirical treatment in COVID-19 and are associated with undesirable consequences, including adverse events, toxicity, resistance and clostridioides difficile infections, it is prudent for physicians to prescribe them judiciously.

At the international level, the scientific community identifies three main lines on which to move. The first is a cultural change leading to the truly appropriate use of antibiotics to reduce their abuse. The second is a strategy aimed at the introduction of innovative therapies capable of dealing with resistant strains. The third raises the importance of vaccines. Indeed, the use of vaccines would reduce the need for antibiotics and help combat the increase in infections with drug-resistant bacteria [41].

The results of the present study should be interpreted with caution due to the limited sample size that reduces statistical power and the predominantly Italian population that may limit the generalizability. Considering the number of patients included in the study, we believe that this is a representative cohort of COVID-19 in Italy. 

However, the study has limitations. Some variables important in determining whether the prescription was appropriate (such as antimicrobial spectrum, start time, cycle duration, etc.) were not available. We were unable to analyze separately the courses of antibiotics started at the time of hospital admission from those started thereafter. Concomitant treatments, especially immunomodulators that may have predisposed patients to co-infections, were not considered. Although we have measured and adjusted for the variables that have so far been shown to be correlated with increased mortality (age and comorbidity), it is possible that other unassessed variables may have been involved. Furthermore, our study is observational in nature, which prevents us from determining causal relationships.

## 4. Materials and Methods

We performed a retrospective observational cohort study on all patients admitted for COVID-19 (discharged and deceased) during the years 2020, 2021 and 2022 at the Hospital Clinic of Ascoli Piceno and San Benedetto del Tronto.

The severity of the disease was defined by the clinician as mild (normal oxygen saturation and normal chest X-ray), medium (radiological evidence of COVID-19 pneumonia) or severe (oxygen saturation ≤ 93%, bilateral lung infiltrates).

Bacterial co-infection was diagnosed based on positive blood, sputum, urine or tissue culture results. Bacterial co-infection was diagnosed on admission or within 48–72 h of hospitalization.

Demographic and clinical factors, including age, gender, comorbidities, disease severity, and bacterial co-infection were extracted from electronic medical records and entered into an Excel spreadsheet in anonymous form. We included patients in both intensive care and general wards. Bacterial co-infection was determined by the presence of characteristic clinical features. Patterns of antibiotic use were also tabulated. 

The collected data were examined using two types of statistical analysis: a univariate and multivariate analysis. First, a descriptive analysis was performed to provide a summary representation of the results of the observations. For continuous variables, the indices used are central tendency indices, such as the mean, and dispersion indices, such as the standard deviation (SD). For categorical variables, absolute and relative frequencies were reported. The t-test was used to compare the averages between groups. Multivariate logistic regression was used to analyze the association between death and the other variables under observation (co-infection, antibiotic use, disease severity). The association between categorical variables was investigated using the χ^2^ linkage test. Results with *p* < 0.05 were considered statistically significant. All analyses were performed using RStudio.

## 5. Conclusions

The rational use of antibiotics requires culture results and sensitivity tests. Where the above-mentioned tests could not be performed, therapy must be guided by epidemiological studies. Are these considerations made in the COVID-19 context?

There are few studies in the literature investigating the impact of antibiotic prescription on an outcome such as death and over such a long-time span. Overall, our results suggest that where secondary bacterial infection is absent, the prescription of antibiotics may not be beneficial for the outcomes of patients with COVID-19 but on the contrary lead to increased mortality especially in more severe patients. In future studies, the implementation of antimicrobial stewardship measures and the use of antibiotics in COVID-19 should lead to careful evaluation, considering the low confirmed rates of bacterial infection.

## Figures and Tables

**Figure 1 antibiotics-12-00912-f001:**
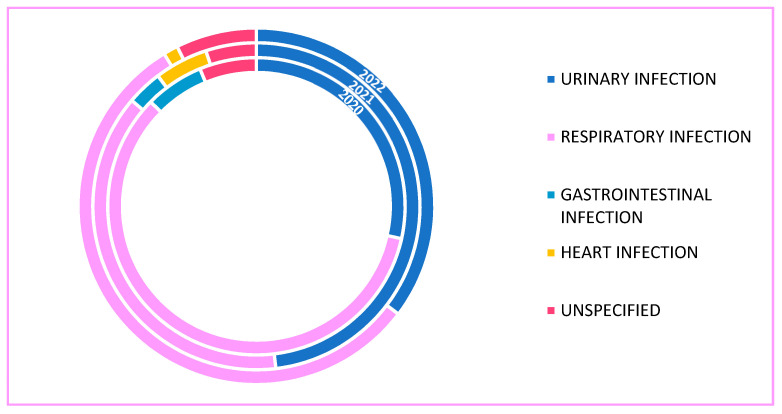
Bacterial co-infection.

**Figure 2 antibiotics-12-00912-f002:**
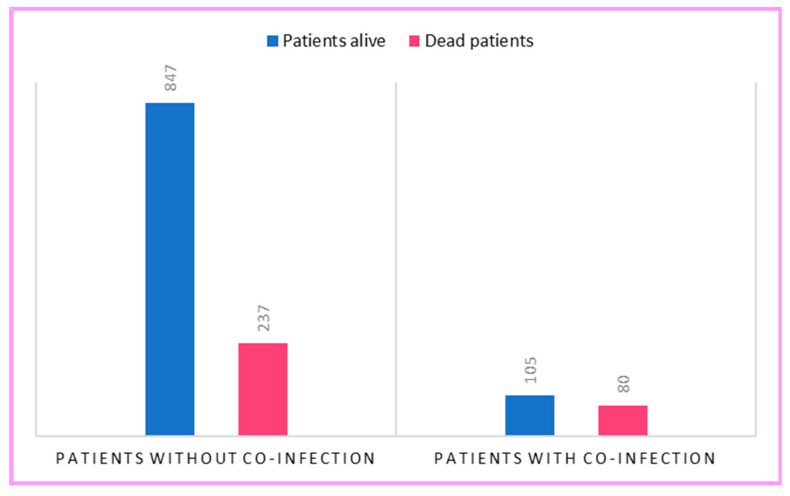
Significant association between death and bacterial co-infection.

**Figure 3 antibiotics-12-00912-f003:**
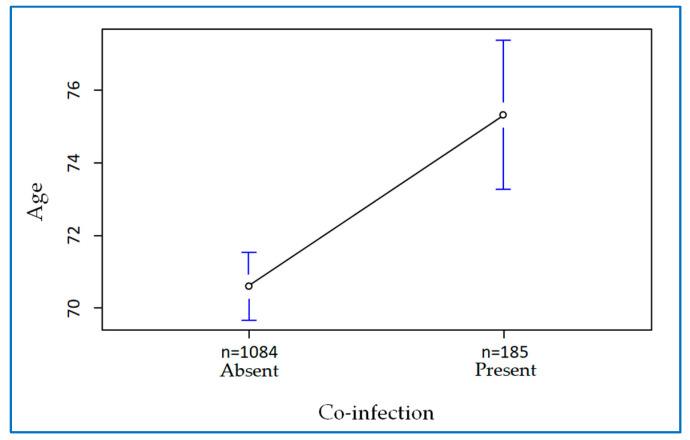
The 95% confidence interval for averages.

**Figure 4 antibiotics-12-00912-f004:**
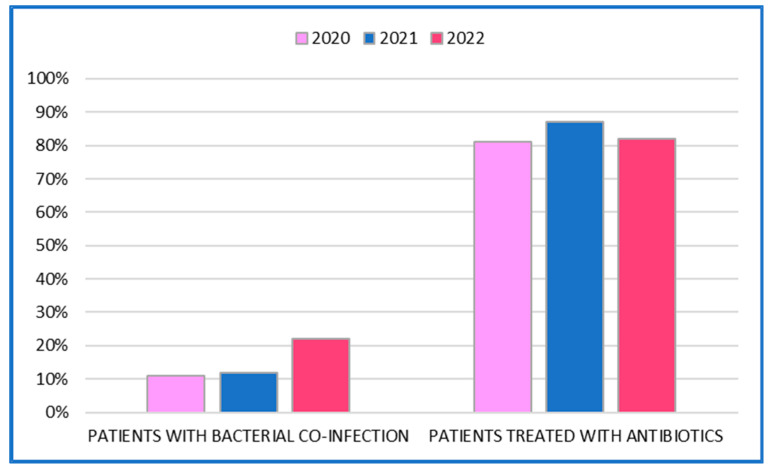
Bacterial co-infection versus antibiotic treatment.

**Figure 5 antibiotics-12-00912-f005:**
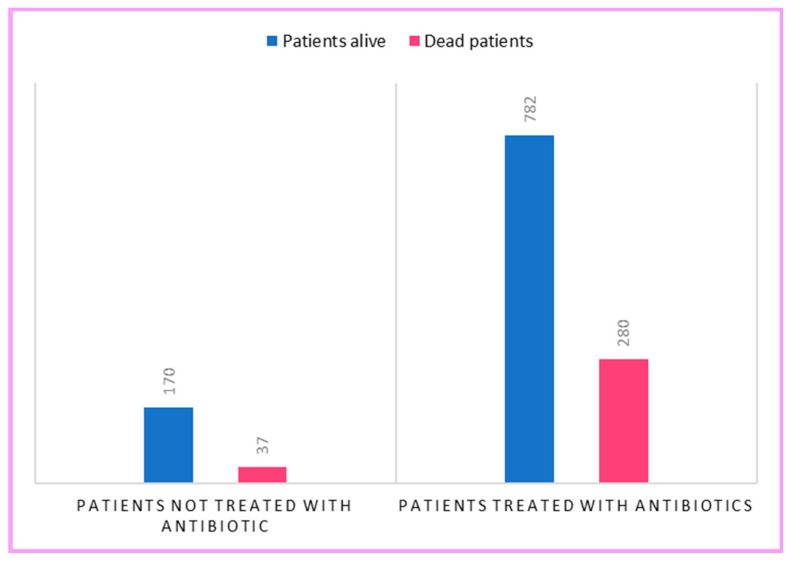
Significant association between death and antibiotic use.

**Figure 6 antibiotics-12-00912-f006:**
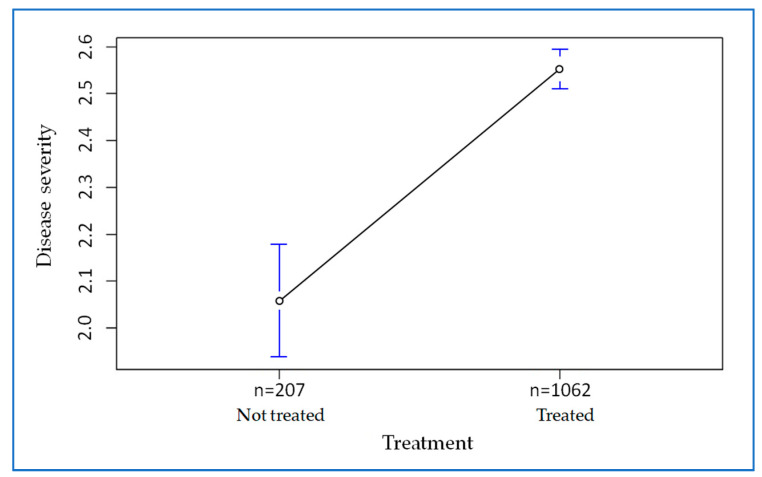
Association between COVID-19 disease severity and antibiotic use.

**Figure 7 antibiotics-12-00912-f007:**
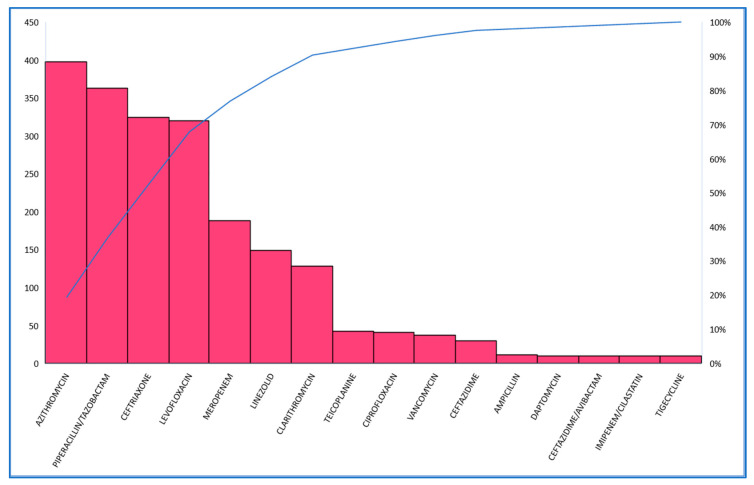
Antibiotics used in the hospitalized patient for COVID-19.

**Figure 8 antibiotics-12-00912-f008:**
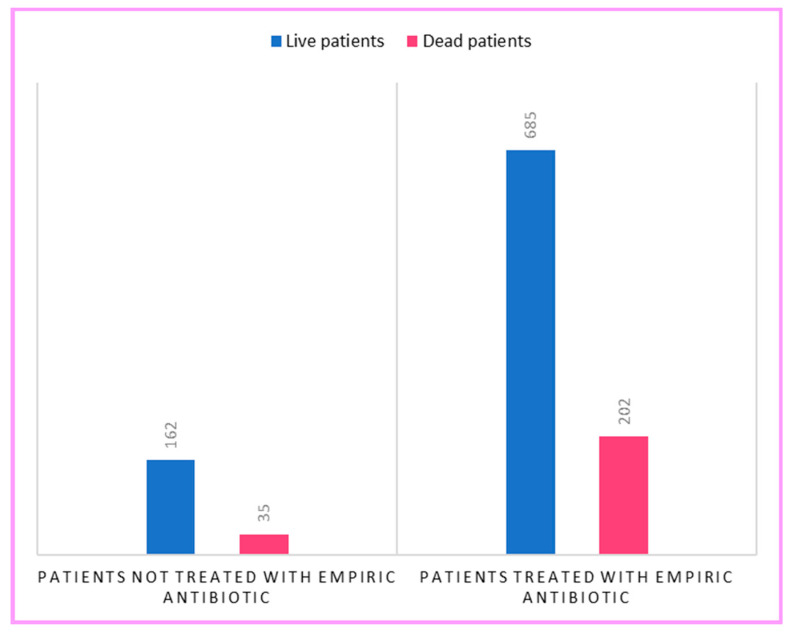
Association between death and empirical antibiotic use.

**Table 1 antibiotics-12-00912-t001:** General characteristics of the sample under analysis.

	February 2020–December 2020	January 2021–December 2021	January 2022–December 2022
** Total (%) **	421 (100)	475 (100)	373 (100)
** Average age (SD) **	70.7 (14.4)	68 (14.7)	76 (16)
** Males (%) **	244 (58)	297 (62.5)	171 (45.8)
** Females (%) **	117 (42)	178 (37.4)	202 (54.2)
** Asthma (%) **	3 (0.7)	2 (0.4)	6 (1.6)
** COPD (%) **	20 (4.7)	11 (2.3)	9 (2.4)
** Obesity (%) **	22 (5.2)	34 (7.1)	22 (5.9)
** Diabetes (%) **	60 (14.3)	91 (19.1)	56 (15)
** Renal failure (%) **	6 (1.4)	8 (1.7)	44 (11.8)
** Cancer pathology (%) **	10 (2.4)	7 (1.5)	34 (9.1)
** Hypertension (%) **	63 (15)	81 (17)	29 (7.7)
** Atrial fibrillation (%) **	25 (6)	37 (7.8)	53 (14.2)
** Ischemic heart disease (%) **	19 (4.5)	25 (5.3)	37 (9.9)
** Heart failure (%) **	31 (7.4)	24 (5)	15 (4)
** Hospitalization for COVID-19 symptomatic (%) **	28 (6.7)	34 (7.2)	132 (35.4)
** Hospitalization for COVID-19 pneumonia (%) **	113 (26.8)	123 (25.8)	47 (12.6)
** Hospitalization for ARDS by COVID-19 (%) **	280 (66.5)	318 (67)	194 (52)
** Average days of hospitalization (SD) **	13.2 (11.4)	13.1 (12.5)	15.2 (14)
** Deaths (%) **	132 (31.3)	106 (22.3)	79 (21.2)

SD—standard deviation, COPD—chronic obstructive pulmonary disease, ARDS—respiratory distress syndrome.

**Table 2 antibiotics-12-00912-t002:** Multivariate logistic regression on factors associated with death.

	β	Standard Error	z	*p*
** Age **	0.083	0.007	11.582	0.000
** Comorbidity **	0.606	0.158	3.828	0.000
** Co-infection **	1.221	0.194	6.281	0.000
** Severity of COVID-19 **	0.783	0.112	6.569	0.000
** Admission department **	0.930	0.170	5.466	0.000

**Table 3 antibiotics-12-00912-t003:** Multivariate logistic regression on factors associated with death.

	β	Standard Error	z	*p*
**Age**	0.062	0.006	10.144	0.001
**Comorbidity**	0.645	0.152	4.221	0.001
** Co-infection **	0.850	0.181	4.698	0.001
** Antibiotic use **	0.493	0.210	2.343	0.019

**Table 4 antibiotics-12-00912-t004:** Multivariate logistic regression on factors associated with death.

	β	Standard Error	z	* p *
** Age **	0.071	0.006	10.664	0.001
** Comorbidity **	0.597	0.157	3.804	0.001
** Co-infection **	0.620	0.188	3.286	0.001
** Azithromycin **	0.121	0.162	0.746	0.455
** Piperacillin/Tazobactam **	0.556	0.158	3.503	0.001
** Ceftriaxone **	−0.046	0.137	−0.339	0.734
** Levofloxacin **	−0.201	0.173	−1.160	0.246
** Meropenem **	0.766	0.211	3.627	0.001
** Linezolid **	0.903	0.237	3.814	0.001

## Data Availability

Data and material are available from the corresponding author.

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
