# Peer review of "Epidemiological Impact on Use of Antibiotics in Patients Hospitalized for COVID-19: A Retrospective Cohort Study in Italy"

_antibiotics, 2023, doi:10.3390/antibiotics12050912_

Round 1

Reviewer 1 Report

The current manuscript titled: "Epidemiological Impact on Use of Antibiotics in Patients Hospitalized for COVID-19: A Retrospective Cohort Study in Italy" represents an important analysis of evolving field of Internal Medicine and Infectious Diseases.

In my opinion, these are the adjustments which should be made to increase the value of your manuscript:

1.      Line 15: change please “COVID 2019” to “COVID-19”.

2.      Line 122: under Table 1, please, add abbreviations.

3.      It is not clear for what purpose the Authors divided the patients’ groups by years. Since, for example, 2020 included 2 COVID-19 variants. It is recommended to categorize patients by COVID-19 successive pandemic waves (Alpha, Beta, and Delta variants) so that there is clinical relevance and the ability to compare these periods.

4.      In the Discussion section, there is not enough comparative information with other studies. Also, it is recommended to study this recent manuscript https://doi.org/10.3390/jpm12060877.

5.      The Conclusions section is very extensive and contains a lot of information that belongs to the Discussions section. Please short the Conclusions section in which indicate the practical implications of your study and its relevance to clinical practice.

6.      Please, add study limitations.

7.      The manuscript contains some punctuation errors, please revise the text (e.g., lines 149, 171, 173, 174, 226, 251, etc.).

8.      Adapt the references according to the Journal requirements.

Minor editing of English language required

Reviewer 2 Report

This is a useful study.

This is a retrospective study about bacterial co-infection rates and empirical antibiotic use in hospitalized Covid-19 patients. 

The topic is relevant in the field and has important results. 

It contains data on empirical antibiotic use in Italy during the Covid-19 era. 

Determination of bacterial co-infection should be explained clearly. 

The conclusions are consistent with the evidence but it can be discussed better. 

The references are appropriate. 

The tables and figures are suitable.  

Reviewer 3 Report

Dear authors,

You sought to explore a topic of great interest, such as the incidence of bacterial co-infections in hospitalized COVID-19 patients and antibiotic consumption over a 3-year period. Unfortunately, the submitted manuscript is subject to several omissions and methodological errors.

1.       Methods are not well written and study’s exploratory endpoint not clearly presented.

2.       Lines 316-317:  “Bacterial co-infection was determined by the presence of characteristic clinical features”.  

Based on which “clinical features” was the diagnosis made? Were they microbiologically documented as mentioned in line 249 (Of these, 83.5% had no laboratory evidence of bacterial infection) ?

3.       The time of co-infection diagnosis is not mentioned in the manuscript. Did they occur earlier or later during hospitalization? Were they more common in patients with severe and critical COVID-19? Did patients’ comorbidities contributed to the occurrence of co-infection?

4.       Study population consisted of patients hospitalized both in medical wards and ICU, but there was not a distinguishment between the 2 groups either in the manuscript or the analyses performed. As easily assumed, critically ill and intubated patients are vulnerable to secondary infections and as a result, greater mortality rates are recorded in this subset of patients.

5.       Did you record the use of immunomodulatory agents (e.g corticosteroids, IL-1b and IL-6 blockade, etc) that predispose to co-infections? Were the co-infections more common in patients that were administered immunomodulatory agents? 

6.       Line 135: “As shown in Figure 2, the bacterial co-infection rate was stable at 11% during the years 2020-2021 with an increase of 10% in 2022.”

How do you interpret this finding? How was it possible an increase of 10% in the co-infection rate in 2022, since COVID-19 severity has been decreased in the era of omicron-variant predominance and they are recommendation against the unnecessary use of antibiotics for the treatment of these patients? Despite the fact that COVID-19 morbidity and hospitalization rates gradually decrease, the incidence of co-infection increases?

7.       Line 223-226: As shown in Figure 7, the proportion of deaths among those who did not receive empirical antibiotics was 17.8% and increased to 22.8% when empirical antibiotics were used. However, the 5% increase found in the 225 sample was not statistically significant (χ2= 2.0815 ; df=1 ; p= 0.1).

How do you interpret this finding? Was the empirical antimicrobial therapy most easily administered in severely ill patients irrespectively of the documentation if co-infection?

8.       Lines 170-172: a higher mortality rate emerged in antibiotic-treated patients than in untreated ones (χ2= 6.22 ; df=1 ; p=0.012). In fact, death occurred in 26.4% of treated patients compared to 17.9% of untreated patients.

How do you interpret this finding?  May antibiotic-allocated patients have experienced either documented co-infection or critical disease and as a result mortality was increased?

9.       Line 163-165: The mean age of co-infected 163 patients was 70.6 years, whereas the mean age of co-infected patients was 75.3  

What do you mean?

10.   Your findings should be better analyzed in “Discussion” and extensive editing of English language is required.

Extensive editing of English language required

Round 2

Reviewer 1 Report

I agree with the changes made by the authors, which greatly improved the quality of the manuscript.

Minor editing of English language required